# Endophytic Bacteria from the Desiccation-Tolerant Plant *Selaginella lepidophylla* and Their Potential as Plant Growth-Promoting Microorganisms

**DOI:** 10.3390/microorganisms12122654

**Published:** 2024-12-21

**Authors:** Maria Guadalupe Castillo-Texta, José Augusto Ramírez-Trujillo, Edgar Dantán-González, Mario Ramírez-Yáñez, Ramón Suárez-Rodríguez

**Affiliations:** 1Laboratorio de Fisiología Molecular de Plantas, Centro de Investigación en Biotecnología, Facultad de Ciencias Biológicas, Universidad Autónoma del Estado de Morelos, Av. Universidad No. 1001, Col Chamilpa, Cuernavaca 62209, Mexico; maria.castillotexta@gmail.com; 2Laboratorio de Fisiología Molecular de Plantas, Centro de Investigación en Biotecnología, Universidad Autónoma del Estado de Morelos, Av. Universidad No. 1001, Col Chamilpa, Cuernavaca 62209, Mexico; augusto.ramirez@uaem.mx; 3Laboratorio de Estudios Ecogenómicos, Centro de Investigación en Biotecnología, Universidad Autónoma del Estado de Morelos, Av. Universidad No. 1001, Col Chamilpa, Cuernavaca 62209, Mexico; edantan@uaem.mx; 4Programa de Genómica Funcional de Eucariontes, Centro de Ciencias Genómicas, Universidad Nacional Autónoma de México, Av. Universidad No. 1001, Col Chamilpa, Cuernavaca 62210, Mexico; mario@ccg.unam.mx

**Keywords:** *Selaginella lepidophylla*, desiccation, endophytes, trehalose, plant growth promotion, resurrection plant

## Abstract

Bacteria associated with plants, whether rhizospheric, epiphytic, or endophytic, play a crucial role in plant productivity and health by promoting growth through complex mechanisms known as plant growth promoters. This study aimed to isolate, characterize, identify, and evaluate the potential of endophytic bacteria from the resurrection plant *Selaginella lepidophylla* in enhancing plant growth, using *Arabidopsis thaliana* ecotype Col. 0 as a model system. Plant growth-promotion parameters were assessed on the bacterial isolates; this assessment included the quantification of indole-3-acetic acid, phosphate solubilization, and biological nitrogen fixation, a trehalose quantification, and the siderophore production from 163 endophytic bacteria isolated from *S. lepidophylla*. The bacterial genera identified included *Agrobacterium*, *Burkholderia*, *Curtobacterium*, *Enterobacter*, *Erwinia*, *Pantoea*, *Pseudomonas*, and *Rhizobium*. The plant growth promotion in *A. thaliana* was evaluated both in Murashige Skoog medium, agar-water, and direct seed inoculation. The results showed that the bacterial isolates enhanced primary root elongation and lateral root and root hair development, and increased the fresh and dry biomass. Notably, three isolates promoted early flowering in *A. thaliana*. Based on these findings, we propose the *S. lepidophylla* bacterial isolates as ideal candidates for promoting growth in other agriculturally important plants.

## 1. Introduction

Desiccation is a phenomenon that causes irreversible damage to the growth, development, productivity, and adaptability of plants; however, some plants have the ability to tolerate stress by resisting or evading negative environmental stimuli or can remain in a state where their phenotype is not drastically affected. Plants face stress through morphological, physiological, metabolic, and genetic changes, although the intervention of certain microorganisms can turn out to be a strategy to counteract stress [1,2]. The *Selaginella* genus is home to around 700 species that inhabit different climates such as desert, temperate, tropical, and arctic areas. Some members of this genus are tolerant to freezing and drought, such as *Selaginella lepidophylla*, which was the first vascular plant reported with the ability to survive a 95% loss of water content and 0–50% relative humidity. It has been suggested that this tolerance is due to the complex cellular structure that maintains the integrity of the material contained in the cell and its high trehalose content [3,4,5,6,7,8]. As mentioned, its ability to tolerate desiccation is attributed to the high accumulation of trehalose, which is independent of the physiological state of the plant (hydrated or dehydrated); however, in other desiccation-tolerant plants, the trehalose content is minimal compared to the total accumulated sugar content. To identify the genes involved in trehalose biosynthesis, such as *TPS1* and *TPS2* in *S. lepidophylla*, many genes that encode *TPS* of microbial origin have been identified, which were isolated from a screening of a cDNA library [8,9]. Furthermore, when grown under sterile conditions, endophyte-free *S. lepidophylla* is unable to regenerate, so it is suggested that the high levels of trehalose contained in the plant, as well as its normal growth and development, are due to endophytes [10]. In *S. lepidophylla*, trehalose represents 90% of the soluble carbohydrates, being the organism with the highest content [4,9,11,12,13]. Microorganisms associated with plants, whether rhizospheric, epiphytic or endophytic, result in a vital interaction for agriculture, particularly endophytic microorganisms, which play an elemental role in the productivity and health of plants. Endophytes colonize plant tissues by entering through the root, flowers, stems, leaves, fruits, cotyledons, stomata, seeds, or wounds, without causing apparent damage but with a benefit for the host plant. Plant growth-promoting bacteria (PGPB) are a heterogeneous group of free-living soil bacteria capable of surviving and colonizing the plant rhizosphere to enhance growth, development, yield, and adaptability. They present direct or indirect mechanisms for the promotion of the growth of the plant (PGP). The mechanisms of action with a direct effect involve plant–microorganism interactions such as the synthesis of growth-stimulating substances (phytohormones) such as indole-3-acetic acid (IAA), cytokinins (CKs), and gibberellins (GAs). Nutrient assimilation and absorption include the ability to produce or change the concentration of growth regulators, biological nitrogen fixation, and solubilization of inorganic phosphates, minerals, and other nutrients. Indirect effects are related to biological control, involving the interaction of the microorganism of interest with a phytopathogen to reduce or prevent harmful effects through competition for an ecological niche or substrate; the synthesis of chemical compounds such as siderophore inhibitors, antimicrobial substances, compounds antifungals, and lytic and detoxifying enzymes; and induced systemic resistance. The latter induces the resistance of systemic tissues to attack from phytopathogens through the emission of volatile organic compounds (jasmonic acid, JA; salicylic acid, SA; ethylene, ET) that participate in plant protection. Various genera have a PGP capacity, including *Agrobacterium*, *Alcaligens*, *Arthrobacter*, *Azospirillum*, *Azotobacter*, *Bacillus Beijerinckia*, *Burkholderia*, *Derxia*, *Enterobacter*, *Herbaspirillum*, *Klebsiella*, *Pseudomonas*, *Rhizobium*, *Rhodopseudomonas*, and *Serratia* [14,15,16]. The wide diversity of PGPB in agricultural systems suggests the elemental role they play in plant physiology and functioning. Studies have shown how beneficial they are in agricultural crops in relation to their growth, development, and adaptability to biotic and abiotic stress due to the diversity of their PGP mechanisms. However, more research needs to be conducted on the various PGPB that produce synergistic effects and their application for PGP.

This study aims to isolate, characterize, and identify the endophytic bacteria of *S. lepidophylla* from its natural environment during the rainy and dry seasons. In addition, we aim to select the ideal candidates to evaluate the promotion of plant growth in *Arabidopsis thaliana* plants with the endosymbionts of *S. lepidophylla*. This will allow us to determine the role of the endophytic microbiota associated with the resurrection plant *S. lepidophylla* to try to understand the role or physiological and molecular contribution of endophytic bacteria to the plant.

## 2. Materials and Methods

### 2.1. Plant Material

Five specimens of *S. lepidophylla* were collected in the town of San Andrés de la Cal in the municipality of Tepoztlán, Morelos, Mexico (18°57′22″ N, 99°06′50″ W, 1495 M.A.S.L.) during the rainy (hydrated state) and drought (dehydrated state) seasons.

### 2.2. Plant Material Sterilization and Bacterial Endophytes Isolation

Excess soil was removed with plenty of tap water; then, 1 g of microphyll and rhizophore were weighed separately. The plant tissues were placed in sterile bottles for disinfection in a laminar flow hood. For the plant tissue in the hydrated state, 10% commercial chlorine (Cloralex^®^, 6.15% sodium hypochlorite, Mexico City, Mexico) was used and 0.05% Triton X-100 was used for 10 min. For the dried plant tissue, commercial chlorine at 20% and Triton X-100 at 0.1% for 20 min were used. Both types of plant tissue were rinsed with sterile distilled water and moisture excess was removed with sterile filter paper. To corroborate the disinfection, the plant tissues were printed in culture media and incubated at 30 °C for 72 h. To isolate endophytic bacteria, the plant tissue was ground with a sterile mortar and pestle by adding 9 mL of 10 mM magnesium sulfate (MgSO_4_). Subsequently, serial dilutions were made, considering the initial dilution as 10^−1^, and these were plated until the 10^−3^ dilution. From each dilution of the rhizophore and microphyll extracts, 100 µL was seeded separately in the four-culture media of nutrient agar (AN), Lysogenic Agar (LA), peptone yeast extract (PY supplemented with CaCl_2_ 7 mM), and potato dextrose agar (PDA). The plates were incubated at 30 °C until the appearance of colonies. After incubation, colonies with different morphology, size, color, and Gram staining were selected. Selected colonies were streaked at least three times and, once purified, they were stored in 20% glycerol (*v*/*v*) at −80 °C for future use.

### 2.3. DNA Extraction and Analysis of Enterobacterial Repetitive Intergenic Consensus Sequences (ERIC-PCR)

DNA was purified with the Puregene^®^ kit (*Qiagen*, Hilden, Germany) following the manufacturer’s instructions for Gram-negative bacteria. The PCR reaction was carried out in a thermocycler under the following conditions: initial denaturation of 1 cycle at 94 °C for 3 min; 35 cycles with denaturation at 94 °C for 1 min, alignment at 46 °C for 90 s and elongation at 72 °C for 2 min; and a final extension step of 72 °C for 8 min. The products were visualized in 1.2% agarose gels. The oligonucleotides used for ERIC-PCR were ERIC-1R 5’-ATGTAAGCTCCTGGGGATTCAC-3’ and ERIC-2 5’-AAGTAAGTGACTGGGGTGAGCG-3’ [17,18]. The endophytic bacteria were grouped according to their morphological characteristics and by comparing the positions of band lanes in each agarose gel.

### 2.4. Molecular Identification of Endophytic Bacteria

For molecular identification of endophytic bacteria, the 16S rRNA gene was amplified using PCR with the primer pair 63F 5′-CAGGCCTAACACATGCAAGTC-3′ and L1401R 5′-CGGTGTGTACAAGACCC-3′ [19,20,21] with the following program: initial denaturation at 95 °C for 5 min; followed by 37 cycles at 95 °C for 50 s, 59 °C for 50 s, and 72 °C for 1.5 min; and a final extension at 72 °C for 10 min. The amplification products were purified with the GeneJet Gel Extraction Kit (Thermo Fisher Scientific, Waltham, MA, USA) following the manufacturer’s instructions. The PCR products were sequenced at Macrogen, Korea. The sequences were analyzed with the BioEdit Sequence Alignment Editor program and subsequently compared with the NCBI GenBank database using a nucleotide BLAST (https://blast.ncbi.nlm.nih.gov/Blast.cgi; accessed on 19 October 2024). [22].

### 2.5. Analysis of Indole Acetic Acid (IAA) Production

Although the Salkowski colorimetric technique is the standard method for determining the ability of bacteria to produce IAA, this technique not only detects IAA but also detects other indoles such as indole pyruvic acid and indole acetamide. To accurately determine IAA production, high-performance liquid chromatography analysis should be used. However, Salkowski’s method is widely used and considered valid [23].

Bacterial isolates were cultured in solid culture media for 48 h at 30 °C, and then inoculated into liquid culture media for 24 h with shaking at 200 rpm overnight at 30 °C. The culture medium was centrifuged at 16,873 rcf for 5 min and the cell pellet was washed with 1 mL of 10 mM MgSO_4_ and centrifuged for 3 min. Finally, it was resuspended in 1 mL of MgSO_4_ and mixed in a vortex to then adjust at 0.1 optical density (OD_600nm_) in the liquid culture medium Jain and Patriquin (JP) without and with tryptophan (Trp; 0.5 g/L) and grown for 24 and 48 h [24]. Once the time had elapsed, 500 µL of the culture was taken, centrifuged, and mixed with 500 µL of the Salkowski reagent (SR) to be incubated for 30 min at room temperature in the dark. The optical density was determined at OD_540nm_ in a single beam UV–visible spectrophotometer model 6405 (190 to 1100 nm), JENWAY [25]. The quantification of IAA production was determined with the SR standard curve with different IAA concentrations (1, 5, 10, 20, and 40 µg) [15,26]. All experiments were repeated three times independently.

### 2.6. Determination of Phosphate Solubilization (PS)

Sample processing was carried out as previously described. The optical density setting (OD_600nm_) was 0.2 to inoculate in 10 µL in Pikovskaya’s (PK) solid culture medium. It was incubated at 30 °C for 7 d, with monitoring on days 1, 2, 6, and 7; on the last day, the diameter of the measurement of the halo indicative of phosphate solubilization was measured [15,27]. For the numerical determination, the following equation was used, expressing the result in millimeters (mm): PSI (phosphate solubilization index) = colony diameter + halo zone diameter/colony diameter [28,29,30,31]. All experiments were repeated three times independently.

### 2.7. Determination of Biological Nitrogen Fixation (BNF)

The bacterial isolates were cultured in their solid culture media for 48 h at 30 °C. Subsequently, a saturated batch was inoculated at the bottom of the test tube with the semisolid NFb culture medium (g/L): malic acid, 5.0; K_2_HPO_4_, 0.5; MgSO_4_.7H_2_O, 0.2; NaCl, 0.1; CaCl_2_. 2H_2_O, 0.02; micronutrient solution (CuSO_4_.5H_2_O, 0.04; ZnSO_4_.7H_2_O, 0.12; H_3_BO_3_, 1.4; Na_2_MoO_4_.2H_2_O, 1.0; and MnSO_4_.H_2_O, 1.175), 2 mL; bromothymol blue (5 g/L in 0.2 N KOH), 1.6 mL; FeEDTA (solution 16.4 gL^−1^), 4 mL; vitamin solution (biotin, 10 mg; pyridoxal-HCl, 20 mg), 1 mL; and KOH, 4.5 g/L. We adjusted the pH to 6.5. A total of 1.6 to 1.8 g agar was used for the semisolid medium [32]. It was monitored for 10 d, observing the growth halo (bacterial film), its height, and the change in color of the culture medium that turned from blue to yellow due to the pH change. The height of the growth zone is relevant because nitrogenase, being sensitive to oxygen, needs to have somewhat anaerobic conditions for its proper functioning [32]. All experiments were repeated three times independently.

### 2.8. Quantification of Trehalose Levels

Sample processing was carried out as described above. To then determine the optical density (OD_600nm_) at 0.1 and inoculate in the liquid minimal culture medium (g/L: succinic acid, 2.5; fructose, 2.5; K_2_HPO_4_, 6.0; KH_2_PO_4_, 4.0; NH_4_Cl, 1.0; MgSO_4_, 0.20, NaCl, 0.10; CaCl_2_, 0.02; FeCl_3_, 0.01; NaMoO_4_, 0.002; KOH, 2.10; adjusted to a pH of 6.0 to 6.2), we supplemented it with 100 mM NaCl and grown for 24 h. After 24 h, the culture was centrifuged, decanted, and stored at −20 °C. Cells were resuspended in 1 mL of 85% absolute ethanol (with HPLC-grade water and absolute ethanol) and incubated at 85 °C for 15 min for cell lysis. It was then centrifuged at 16,873 rcf for 5 min to transfer the supernatant to a 1.5 mL microcentrifuge tube, and incubated at 85 °C with an open lid until total evaporation of ethanol was achieved. The quantification of the trehalose content was carried out using high-performance liquid chromatography (HPLC) with a SUPELCOSIL LC-NH_2_ column (SIGMA-ALDRICH) and mobile-phase acetonitrile/water 75:25 (*v*/*v*) at a flow of 1 mL/m. A standard curve (1.25, 0.625, 0.325, and 0.156 µg/mL) was performed using trehalose (Sigma Chemical Co., St. Louis, MO, USA) [33,34]. This experiment was repeated once independently.

### 2.9. Siderophore Production Assay

The preparation of the culture medium was as described by Louden [35]. Once the Petri dishes were filled, they were inoculated at an OD of 0.2 (OD_600nm_) in 20 µL and incubated at 28 °C for 10 d. The numerical determination was based on the following equation: colony diameter + halo zone diameter/colony diameter, expressed in millimeters [29,31,36]. All experiments were repeated three times independently.

### 2.10. Arabidopsis Thaliana Growth Promotion Assay

Seeds of *A. thaliana* ecotype Col. 0 were rinsed with water and sterilized with 20% NaClO and 0.2% tween 20 for 5 min. Subsequently, they were washed five times with sterile distilled water for 1 min and incubated in 1 mL of KNO_3_ (1500 ppm) for 3 d at 4 °C under continuous light.

To evaluate the effect of the bacterial isolates on the development of *A. thaliana*, 10 seeds were placed per Petri dish with Murashige and Skoog 50% (MS 50%) culture medium [37] and agar water (AW; 10 g/L agar; pH 5.9). The Petri dishes were placed vertically at an angle of 65° with a photoperiod of 16 h light/8 h dark at 25 °C for 10 d to allow the growth of the aerial part and the root along the agar surface. Subsequently, from 2 cm from the root tips (Figure 1), the bacterial isolates were inoculated at an OD_600nm_ of 0.2. The plant growth-promotion effect was determined 10 days post-inoculum, considering the plant length which was measured with a digital caliper, the characteristics of the main root and lateral roots visualized in a stereoscopic microscope (Nikon SMZ1500), as well as biomass in fresh weight and dry weight. The strains *Azospirillum brasilense* Cd and *Arthrobacter chlorophenolicus* 30.16 were used as positive control bacteria. All experiments were repeated three times independently (n = 30).

To evaluate the effect of bacterial isolates on germination, direct inoculation was carried out on *A. thaliana* seeds. In this experiment, 21 bacterial isolates were used (13 from hydrated and 8 from dehydrated state), which showed the best results in the experiment in MS 50% and AW culture medium. The protocol to follow was as described previously: once they were incubated in KNO_3_, the seeds were inoculated with the bacterial isolates for 10 min, and then they were placed in Petri dishes with AW culture medium with a photoperiod of 16 h light/8 h of darkness at 25 °C for 10 d. The average daily germination (GDM), maximum germination value (MV), germination percentage (PG), and value of germination vigor or germination speed (VG) were determined. Additionally, on day 10, the length of the plant, its characteristics, and biomass in fresh and dry weight were evaluated. All experiments were repeated three times independently (n = 50).
GDM = germination percentage per day between days

MV = final germination percentage divided by the number of days

PG = seeds germinated (SG) among seeds sown (SS) per 100: SG/SS × 100

VG = MV × GDM

### 2.11. Statistical Analysis

All experiments were repeated at least three times independently unless indicated. The data were processed using analysis of variance (ANOVA) followed by a post-hoc test to compare means with a Tukey multiple range test. The alpha value (α) was 0.05 and the value Pr > F, which is the probability of the F value of the model and indicates whether a calculated F value is significant or not based on the benchmarks of 0.05 and 0.01 generally used in research. The SAS program (Statistical Analysis Software 9.0) was used for statistical analysis.

## 3. Results

### 3.1. Endophytic Bacteria Isolated from Selaginella lepidophylla

A total of 163 endophytic bacteria of *S. lepidophylla* were obtained. These were obtained from two collection conditions (hydrated and dehydrated), two plant tissues (microphyll and rhizophore), and four different growth media (AN, LB, PY, and PDA). For the rainy season (hydrated state), 122 isolates (74.85%) were obtained, and in the dry season (dehydrated state), 41 isolates (25.15%) were obtained. Regarding the plant tissue, the microphyll obtained 87 isolates (53.37%) and the rhizophore obtained 76 isolates (46.63%) (Table 1).

### 3.2. Analysis of Enterobacterial Repetitive Intergenic Consensus Sequences (ERIC-PCR)

The ERIC-PCR analysis allowed us to group our 163 bacterial isolates into 114 different banding patterns, thus reducing the number of bacteria which were later used to determine the growth promotion parameters. Within the 114 banding patterns obtained, 28 of these patterns grouped 77 endophytic bacteria which had similarities in their banding patterns, while the rest of the patterns were unique (86 patterns).

### 3.3. Molecular Identification of Endophytic Bacteria

For molecular identification, 65 endophytic bacterial isolates were selected, which obtained the best results in the plant growth-promotion parameters (Appendix A). Forty-nine bacterial isolates collected during the rainy season (hydrated) and sixteen isolates collected during the drought season (dehydrated) were selected. The bacterial genera found for both collection periods were *Pseudomonas* (58.46%), *Pantoea* (21.54%), *Rhizobium* (4.61%), *Curtobacterium* (4.61%), *Agrobacterium* (3.08%), *Streptomyces* (3.08%), and *Burkholderia*, *Enterobacter*, and *Erwinia* (1.54% each). In the rainy season, the genera that predominated were *Pseudomonas* (63.27%), *Pantoea* (20.41%), *Curtobacterium* (6.12%), *Rhizobium* (4.08%), *Streptomyces* (4.08%), and *Burkholderia* (2.04%). The genera *Curtobactecrium* and *Burkholderia* were only found in the rainy season. During the drought season, the predominant genera were *Pseudomonas* (43.75%), *Pantoea* (25%), and *Agrobacterium* (12.5%), followed by *Rhizobium*, *Enterobacter*, and *Erwinia* (6.25% each); these last three genera were only found in the drought season. The accession numbers of the 16S rRNA gene sequence deposited in the NCBI GenBank are found in Appendix A.

### 3.4. Indole-3-Acetic Acid Production (IAA)

In this experiment, the 163 bacterial isolates were used in the JP culture medium without the addition of Trp. Only one IAA producer was obtained at 48 h for the rainy season: *Pseudomonas* sp. SlL116 with 1.62 µg/mL, corresponding to 0.81%. In the medium supplemented with Trp, for the rainy season isolates, 82.78% (101 isolates) produced IAA with a range of 0.037 to 55.13 µg/mL, the latter corresponding to *Pantoea* sp. SlL44. For the isolates in the dry season, 73.17% (30 isolates) were obtained as producers of IAA with a range of 0.71 to 50.73 µg/mL, the latter corresponding to *Enterobacter* sp. SlS9. Based on the production ranges, bacterial isolates were classified into low, medium, and high IAA producers (Table 2). It is worth mentioning that isolates SlL73 and SlL112 only underwent a determination of IAA production, because they subsequently stopped growing. It is worth mentioning that 13 isolates (10 from the hydrated and 3 from the dehydrated stage) were consumers of the IAA produced at 24 h. The IAA consumption values were 12.64, 10.82, 8.09, 7.28, 7.12, 3.72, 2.92, 2.63, 2.57, 1.89, 1.65, and 1.28 μg/mL for isolates SlS19, SlL103, SlL53, SlL57, SlS25, SlL56, *Burkholderia* sp. SlL91, *Pseudomonas* sp. SlL36, SlL86, SlL87, SlS12, and *Pseudomonas* sp. SlL29, respectively. For the remaining two isolates, their consumption was not greater than 0.3 μg/mL, with 0.25 and 0.08 μg/mL for *Pseudomonas* sp. SlL10 and SlL37, respectively. According to the statistical analyses, significant differences were presented, obtaining a value of F = 321.07 and F = 3144.64 for 24 and 48 h, respectively.

### 3.5. Phosphate Solubilization (PS)

After 7 days, the halo indicative of PS was measured, which changed the culture medium from white to transparent and was classified according to what was reported by Milagres et al. [28]. In this experiment, 163 bacterial isolates were used; of those isolated from the rainy season, 57.37% (70 isolates) were phosphate solubilizers, with a range of 2.17 to 3.83 mm, the latter corresponding to SlL121. For the isolates from the drought season, 34.24% (14 isolates) were phosphate solubilizers, with a range of 2.26 to 3.39 mm, the latter corresponding to *Erwinia* sp. SlS1 (Figure 2A).

### 3.6. Biological Nitrogen Fixation (BNF)

This experiment allowed us to qualitatively evaluate which of the 163 bacterial isolates could be nitrogen fixers. In total, 81.59% (133 isolates) were nitrogen fixers, with 100 isolates from the rainy season and 33 from the drought season. For this experiment, *A. brasilense* Cd was used as a positive control.

### 3.7. Trehalose Quantification

A total of 119 bacterial isolates were selected, considering their PGP parameters and the ERIC-PCR analysis, with 97 isolates for the rainy season and 22 for the drought season. For this experiment, we used *A. brasilense* Cd, which presented a content of 2.58 µg/mL, and *A. chlorophenolicus* 30.16, with a content of 48.77 µg/mL, the latter being a natural overproducer of this disaccharide. In the rainy season, 88.65% (86 isolates) of trehalose producers were present, with a range of 0.0038 to 35.75 μg/mL, the latter corresponding to *Pseudomonas* sp. SlL10. In the drought season, 86.36% (19 isolates) of trehalose producers were present, with a range of 0.021 to 19.30 μg/mL, the latter corresponding to *Pseudomonas* sp. SlS4.

According to the information previously detailed for each experiment and contained in Appendix A, which includes the general parameters for the characterization of endophytic bacteria that promote plant growth, we selected candidates for siderophore and PGP production experiments in *A. thaliana* Col. 0. For this, the collection time, trehalose content, IAA production, PS, and unique characteristics of the isolates were considered. We included 63 bacterial isolates for both collection seasons, of which for the rainy season there were 48 bacterial isolates and for the drought season there were 15 bacterial isolates.

### 3.8. Siderophore Production

For the rainy season, 48 bacterial isolates were selected, of which 68.75% (33 isolates) were siderophore producers. In addition, two types of siderophores were identified: hydroxamate, which had an orange color and were the most prevalent, with 87.87% for the rainy season (29 isolates); followed by catecholate, which had a pink color, with 12.12% for the rainy season (4 isolates). For the rainy season, *Pseudomonas* sp. SlL116 was the largest producer of hydroxamate-type siderophores, with 4.60 mm, and *Burkholderia* sp. SlL91 was the largest producer of catecholate type, with 4.04 mm. For the drought season, 15 bacterial isolates were selected, of which 80% (12 isolates) were siderophore producers. For the hydroxamate-type siderophores, 91.67% (11 isolates) were obtained, and for the catecholate type, 8.33% (1 isolates) were obtained. For the drought season, *Pseudomonas* sp. SlS14 was the largest producer of hydroxamate-type siderophores, with 2.15 mm, and *Pseudomonas* sp. SlS39 was the largest and only producer of the catecholate-type siderophores, with 2.40 mm (Figure 2B).

### 3.9. Promotion of Plant Growth in Arabidopsis thaliana

#### 3.9.1. PGP in *A. thaliana* in the MS 50% Culture Medium

For this experiment, 63 previously selected bacterial isolates were used (Table 3). The non-inoculated plants had a size of 74.24 mm with few lateral roots, villi (+++), and large leaves, a fresh weight (FW) of 0.2739 g and a dry weight (DW) of 0.0186 g. The positive control plants, *A. brasilense* Cd, had a size of 72.86 mm with few lateral roots, villi (++), and large leaves, a FW of 0.1966 g and a DW of 0.0171 g; and *A. chlorophenolicus* 30.16 had a size of 81.17 mm with lateral roots (+++), villi (++), and large leaves, a FW of 0.305 g and a DW of 0.0418 g (Table 2). For those isolated from the rainy season, *Rhizobium* sp. SlL9 presented the most prominent size, with 71.74 mm, and thicker lateral roots but few villi. In relation to biomass, *Pseudomonas* sp. SlL106 had the highest FW, with 0.9001 g, and *Pseudomonas* sp. SlL5 had the highest DW of 0.1009 g (Figure 3A,B). For those isolated from the dry season, *Rhizobium* sp. SlS28 presented the most prominent size, with 75.13 mm, with lateral roots (+++), villi (+++), and large leaves. In relation to biomass, *Enterobacter* sp. SlS9 presented the highest FW and DW, with 0.506 g and 0.0933 g (Figure 3C,D). According to the statistical analyses, significant differences were presented, obtaining a value of F = 25.68 and F = 43.07 for rain and drought, respectively. It should be noted that three bacterial isolates (*Enterobacter* sp. SlS9, *Rhizobium* sp. SlS28, and *Burkhoderia* sp. SlL91), as well as the positive control *A. chlorophenolicus* MOR30.16, promoted the early flowering of *A. thaliana* (Figure 4). However, the PGP of the bacterial isolates did not exceed the controls; this could be because the MS 50% culture medium is still rich, and, therefore, both the endophytic bacteria and the plants did not see the need to interact to enhance growth. Therefore, we decided to repeat the test in a culture medium without nutrients, such as an agar water medium.

#### 3.9.2. PGP in *A. thaliana* in the AW Culture Medium

For this experiment, the 63 selected bacterial isolates were used again (Table 3). The non-inoculated plants had a size of 28.04 mm with a primary root that ranged from medium to long, lateral roots (++), and villi (+++), and a FW of 0.01 g and DW of 0.0017 g. The positive control plants, *A. brasilense* Cd, had a size of 16.33 mm, with a primary root that ranged from short to medium, lateral roots (+), and villi (+++), and a FW of 0.0087 g and DW of 0.0004 g, and *A. chlorophenolicus* 30.16 had a size of 12.32 mm with a short primary root, without lateral roots, villi (++) and with chlorosis, FW of 0.0076 g and DW of 0.0014 g. For bacterial isolates in the rainy season, *Pseudomonas* sp. SlL23 had the longest length, with 43.21 mm, and it had lateral roots (++) and villi (++). In relation to biomass, *Pseudomonas* sp. SlL116 had the highest FW of 0.0809 g, and *Pseudomonas* sp. SlL23 presented the highest DW, with 0.0032 g (Figure 5A,B; Table 2). For bacterial isolates in the dry season, *Rhizobium* sp. SlS28 presented the most prominent size, with 42.63 mm, with a medium to long main root, lateral roots (+), and villi (+). In relation to biomass, *Pseudomonas* sp. SlS4 presented the highest FW, with 0.0479 g, and *Pseudomonas* sp. SlS38 had the highest DW, with 0.003 g (Figure 5C,D). According to the statistical analyses, significant differences were presented, obtaining a value of F = 21.36 and F = 16.43 for rainy and drought, respectively. Figure 6 shows the appearance of the *A. thaliana* plants inoculated with the isolates that presented the best results.

#### 3.9.3. PGP in the Germination In Vitro: Direct Inoculation of Bacterial Isolates in *A. thaliana* Seeds in the AW Culture Medium

From the PGP experiments in MS 50% and AW culture medium, 21 bacterial isolates were selected that presented PGP qualities in terms of both plant length and biomass to be evaluated in a direct inoculation on *A. thaliana* seeds (Table 3). The non-inoculated plants had a size of 7.22 mm with a long main root with villi (+++), without lateral roots, and with large and green leaves, a FW of 0.0021 g, and a DW of 0.0010 g. The positive control plants, *A. brasilense* Cd, had a size of 8.30 mm with a medium to long main root with villi (++), without lateral roots and with small and green leaves, a FW of 0.0021 g, and a DW of 0.0001 g. *A. chlorophenolicus* 30.16 had a size of 3.98 mm with a short main root with lateral roots, without villi and with large and green leaves, a FW of 0.0010 g, and a DW of 0.0003 g. Among the 13 bacterial isolates from the rainy season, *Pantoea* sp. SlL67 had the longest length, with 14.35 mm, with a short main root with villi (++) and lateral roots (++) with curved ends (hooks, curls), with medium and green leaves, a FW of 0.0059 g, and a DW of 0.0008. Among the eight bacterial isolates from the drought season, *Rhizobium* sp. SlS28 had the longest length, 16.48 mm, with a medium main root with villi (+), medium and green leaves, a FW of 0.0143 g, and a DW of 0.0017 g (Figure 7A). According to the statistical analyses, significant differences were presented, obtaining a value of F = 8.21 (Pr > F < 0.0001; α = 0.05). In relation to the biomass for the rain isolates, *Pseudomonas* sp. SlL23 presented the highest FW of 0.0122 g, and *Rhizobium* sp. SlL30 had the highest DW, with 0.0012 g. For drought isolates, *Rhizobium* sp. SlS28 had the highest FW of 0.0143 g, and *Pantoea* sp. SlS39 presented the highest DW, with 0.0018 g (Figure 7B). According to the statistical analyses carried out for the FW, significant differences were presented, obtaining a value of F = 6.4 (Pr >F < 0.0001; α = 0.05).

In Figure 7C, the germination percentage on day 1, 4, 7, and 10 is shown, and Figure 7D shows the germination speed on day 10 (F = 2.86, Pr > F 0.0001, α = 0.05). On day 1, *Pseudomonas* sp. SlL116 presented the highest germination percentage (70.45%), while *Pantoea* sp. SlL67, *Pantoea* sp. SlS3, *Pseudomonas* sp. SlS4, *Enterobacter* sp. SlS9, *Pantoea* sp. SlS15, *Rhizobium* sp. SlS28, and *Pantoea* sp. SlS39 had not germinated. The non-inoculated plants presented 55.55%, *A. brasilense* Cd 44.25%, and *A. chlorophenolicus* 30.16 47.22% (F = 24.34, Pr > F < 0.0001, α = 0.05). Figure 7E shows the germination speed of *A. thaliana* seeds inoculated with the bacterial isolates from day 0 to 10. We can highlight in the graph the exact day on which they reached their maximum germination percentage. For the non-inoculated plants and *A. brasilense* Cd this was on day 6, and 93.93% and 80.51% were obtained, respectively. For *A. chlorophenolicus* 30.16 this was on day 4 with 63.88%. *Pantoea* sp. SlL67, *Pantoea* sp. SlS3, *Pantoea* sp. SlS39, and *Enterobacter* sp. SlS9 were the only ones with 100% germination between day 2 and 5. However, *Enterobacter* sp. SlS9 presented a delay in the development and growth of the plant and, therefore, in the biomass. *Streptomyces* sp. SlL20 presented the lowest germination percentage (72.87%) on day 5. For the rest of the bacterial isolates, the germination percentage ranged between 81.39% and 97.61% between days 2 and 10. The non-inoculated plants presented 93.93%, *A. brasilense* Cd 80.51%, and *A. chlorophenolicus* 30.16 63.88% (F = 2.93, Pr > F 0.0008; F = 2.81, Pr > F 0.0013; F = 3.28, Pr > F 0.0003; α = 0.05; day 4, 7, and 10, respectively). Figure 8 shows the appearance of *A. thaliana* seeds inoculated with the bacterial isolates that showed the best results on the elongation and biomass.

## 4. Discussion

The use of endophytic bacteria associated with plants that promote a direct or indirect advantage turns out to be an ideal strategy to obtain an increase in agricultural productivity, reduce the use of chemical fertilizers, and minimize soil and water deterioration and contamination [38,39,40]. The application of PGPB as a biofertilizer is a promising strategy to improve crop efficiency. However, the efficiency of PGPB largely depends on both extrinsic and intrinsic factors, plant genetics, environmental conditions, and other specific characteristics of agricultural systems. Our bacterial isolates were consistent with those reported in other works where PGPR and PGPB were isolated [14,15,16,26,38]. In this work, a considerable microbial collection was obtained, with 163 endophytic bacteria isolated (122 isolates belonged to the rainy season and 41 isolates to the drought season), and we observed that the composition of microbial populations depended on the time of year (rain or drought), plant organ (microphyll and rhizophore), and probably the endophyte species [2,39].

The abundance of endophytic bacteria during the rainy season suggests their probable or future participation through the synthesis, accumulation, degradation, and/or assimilation of compounds that they would be occupying in the dry season. However, having many endophytes does not necessarily mean that they are cooperating with the plant, as some will be present momentarily (taking refuge) and others will prevail. Sixty-five bacterial isolates were selected for molecular identification. The genera identified were *Agrobacterium*, *Burkholderia*, *Curtobacterium*, *Enterobacter*, *Erwinia*, *Pantoea*, *Pseudomonas*, *Rhizobium*, and *Streptomyces*, all of which were previously isolated in other works of PGPB and PGPR and were even isolated from medicinal and drought-tolerant plants [41,42,43]. *Pseudomonas* and *Pantoea* were found most frequently, and both receive special attention because they are bacteria capable of quickly colonizing plant roots, adapting and interacting favorably with other bacteria, in addition to producing compounds, such as phytohormones, secondary metabolites, volatile organic compounds, osmolytes, and polysaccharides, which can contribute to the growth and proliferation of other microorganisms [38,40,41,42,43,44,45,46,47,48].

IAA is a phytohormone responsible for regulating multiple functions in plants; however, it can also bring negative effects due to an imbalance, although the producing bacteria could tolerate and incorporate IAA [14,49,50,51,52,53,54]. According to the literature, around 80% of PGPB and PGPR produce IAA [49,50], which correlates with our results for the isolates in the rainy (82.78%) and drought (73.17%) seasons. In the rainy season, isolates with a low IAA content predominated (62.37%), while in the dry season, those with a high IAA content predominated (63.33%). This could be because during the dry season more isolates with high production are needed as this compound could be collaborating with the desiccation tolerance of *S. lepidophylla*.

Phosphate is involved in processes such as the development of roots, seeds, flowers, membrane phospholipids, and nucleic acids, and it participates in photosynthesis, glycolysis, respiration, and signaling, among others. Previous reports indicate that from 1 to 50% of bacteria are capable of solubilizing phosphates [16,27,52,54,55,56]. We obtained 70 phosphate-solubilizing isolates (57.37%) for the rainy season and 14 isolates (34.14%) for the drought season. *Selaginella* is associated with many phosphate-solubilizing endophytes that show higher solubilization rates than those reported by Castro-González [41] and Fernandez-Júnior [43].

Nitrogen is an elemental molecule due to its participation in the synthesis of nucleic acids, vitamins, proteins, amino acids, and nitrates [32,57]. Various studies have revealed that BNF is related to PGP, such as phosphate solubilization, ACC deaminase activity, IAA biosynthesis, and even biological control and bioremediation [41]. Our results showed that 133 bacterial isolates (81.59%) fixed nitrogen, with 81.96% in the rainy season (100 isolates) and 80.48% (33 isolates) in the dry season.

Endophytic bacteria can produce, secrete, and/or exude polysaccharides that help soil stability to optimize environmental conditions and prevent sudden changes that endanger the plant and its microbiota. The production of compatible solutes, such as trehalose, helps contain this type of stress [16,34,58,59]. We determined trehalose levels in a similar way to García [34], and analyzed 119 bacterial isolates for both periods, of which only 105 isolates produced trehalose. For the rainy season, 88.65% (86 isolates) produced trehalose, with a range of 0.0038 to 35.75 μg/mL, and for drought this was 86.36% (19 isolates), with a range of 0.021 to 19.30 μg/mL. It has been reported that desiccation-tolerant *Pseudomonas* present mechanisms to counteract saline and osmotic stress, including through the accumulation of compatible solutes such as glycine betaine and trehalose, as well as ion transporters that help bacteria prevail and colonize plants from arid soils [60]. In our work, the best producers of trehalose were *Pseudomonas* sp. (SlL10; SlS4).

Iron is elemental for organisms due to its participation as a catalytic center, electron carrier, cofactor, oxygen transport, DNA synthesis and repair, catalyst of enzymatic processes, biofilm formation, BNF, respiration, oxidative phosphorylation, IAA formation, detoxification of free radicals, reduction of nitrates, formation of chlorophyll, photosynthesis, activators of induced systemic resistance, in addition to limiting it for pathogens, thus inhibiting their proliferation [15,16,30,61,62,63]. Plants and bacteria have developed sophisticated mechanisms for the assimilation of this micronutrient through siderophores, which are low molecular weight molecules with a high affinity to Fe^3+^. PGPB can synthesize siderophores and sequester Fe by forming the siderophore-Fe^3+^ complex. This complex is easily assimilated by the microorganisms that synthesized them and those that are nearby, and the concentration of this complex is high enough to nourish the plant. More than 500 siderophores have been identified which have various ligand groups such as hydroxamic acid (hydroxamates; orange color), catechol (catecholates *sensu stricto* formed by 2,3-dihydroxybenzoic acid; pink color), and/or phenolates (composed of SA; pink color), α-hydroxycarboxylic acid (hydroxycarboxylates; one radical is replaced by a double bond with oxygen and nitrogen of the skeleton by a carbon; reddish color), and a mix thereof. In our work over both collection periods, 88.89% of isolates (40 isolates) were producers of hydroxamate-type siderophores and 11.11% (5 isolates) were producers of catechol-type siderophores, which is indicative of the versatility of the bacterial isolates for the production and use of siderophores, which can confer ecological advantages in rhizosphere colonization and plant health.

*A. thaliana* Col. 0 was used to evaluate the PGP of our endophytic bacteria, because this plant has a short life cycle, grows in limited spaces, and presents natural variations (ecotypes), in addition to being counted with various transgenic and mutant lines that can be used in the future to evaluate other capacities of endophytic bacteria. Sixty-five isolates selected for their outstanding characteristics, such as high trehalose content, IAA synthesis, or phosphate solubilization, were used to evaluate their effect when inoculated into *A. thaliana*.

Particularly, in the experiment in MS 50% culture medium, several endophytes were similar in terms of the plant length compared to the controls. However, in relation to biomass, we were able to verify that our endophytes (*Pseudomonas* sp. SlL106, *Pseudomonas* sp. SlL97, *Pantoea* sp. SlL46, *Pseudomonas* sp. SlL104, *Pseudomonas* sp. SlL116, *Pantoea* sp. SlL47, *Pseudomonas* sp. SlL114, and *Burkholderia* sp. SlL91 for the rainy season; and *Enterobacter* sp. SlS9, *Pantoea* sp. SlS27, *Pseudomonas* sp. SlS36, *Pseudomonas* sp. SlS38, and *Rhizobium* sp. SlS28 for the drought season) presented a higher fresh weight and dry weight compared to controls. Furthermore, *Enterobacter* sp. SlS9, *Rhizobium* sp. SlS28, and *Burkhoderia* sp. SlL91, as well as *A. chlorophenolicus* 30.16, induced early flowering in *A. thaliana*. Early flowering can be induced by various biotic and abiotic factors; recent studies have shown that microbial communities are essential because they influence the phenology and flowering of the plant [64].

This study reveals the potential of the desiccation-tolerant plant *S. lepidophylla* to obtain microorganisms with biotechnological potential to promote growth and early flowering, but more studies are necessary to validate its effect on plants of agricultural interest.

## 5. Conclusions

*S. lepidophylla*, being a desiccation-tolerant plant, presents a microbial collection with characteristics that improve plant growth, and could even help crops in arid areas. The wide diversity of endophytic bacteria associated with plants suggests the elemental role they play in their physiology and functioning. We highlighted the biotechnological potential of endophytic bacteria isolated from *S. lepidophylla* in the contribution of promoting plant growth in *A. thaliana* plants, which can be extrapolated to plants of agricultural interest. Our future studies aim to evaluate endophytic bacteria in a crop like *Solanum lycopersicum*, as well as to evaluate its antagonistic activity against bacterial and fungal pathogens in siderophore-producing isolates.

Combinations of endophytic bacteria in an inoculum could give better results in the development, yield, and adaptability of agricultural crops. However, more research is needed on their synergistic effects and their application to promote plant growth. It is worth mentioning that our in vitro experiments and the PGP were evaluated under controlled conditions; in the natural environment, various factors, both biotic and abiotic, can influence plant–endophyte interactions. The complexity of PGP by plant-associated endophytic bacteria needs to be addressed with a comprehensive approach based on in vitro, in vivo, and field studies, as well as evaluating the compatibility of microorganisms in complex communities.

Some limitations of this work are that only the endophytic bacterial microorganisms that were culturable were obtained; likewise, in this work only a limited number of endophytic bacteria were identified. Omics tools such as metagenomics must be used to understand the wide diversity of microorganisms that *Selaginella* harbors.

## Figures and Tables

**Figure 1 microorganisms-12-02654-f001:**
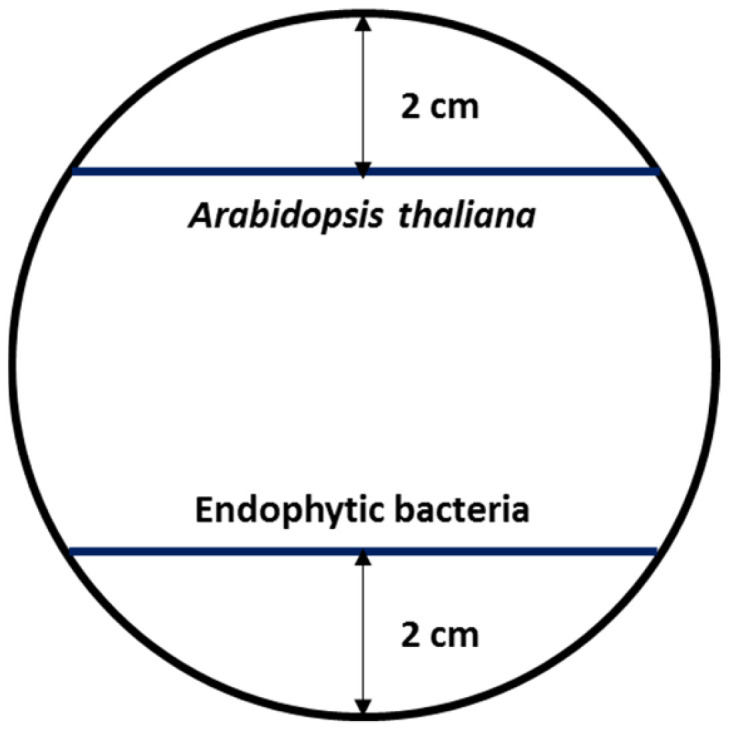
Scheme of inoculation of *Arabidopsis thaliana* plants in MS 50% and AW culture medium with endophytic bacteria.

**Figure 2 microorganisms-12-02654-f002:**
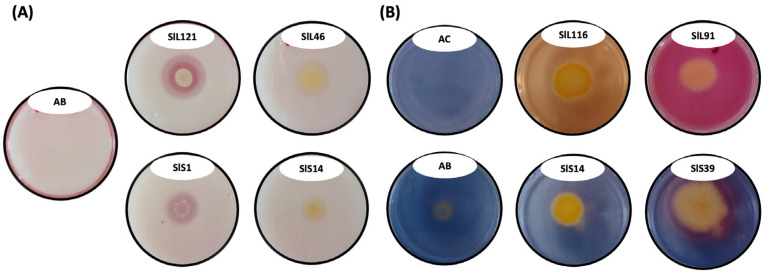
Representative image of the in vitro plant growth-promotion parameters. (**A**) Solubilization of phosphates. AB, *A. brasilense* Cd; *Pseudomonas* sp. SlL121, higher PS for rainy season; *Pantoea* sp. SlL46, lower PS for rainy season; *Erwinia* sp. SlS1, higher PS for drought season; *Pseudomonas* sp. SlS14, lower PS for drought season. (**B**) Production of siderophores. AC, *A. chlorophenolicus* 30.16; AB, *A. brasilense* Cd; *Pseudomonas* sp. SlL121 is a major producer of rainy season hydroxamate-type siderophores; *Burkholderia* sp. SlL91 is the largest producer of rainy season catechol-type siderophores; *Pseudomonas* sp. SlS14 is a major producer of hydroxamate-type siderophores for drought season; *Pseudomonas* sp. SlS1 is a major producer of catechol-type siderophores for drought season.

**Figure 3 microorganisms-12-02654-f003:**
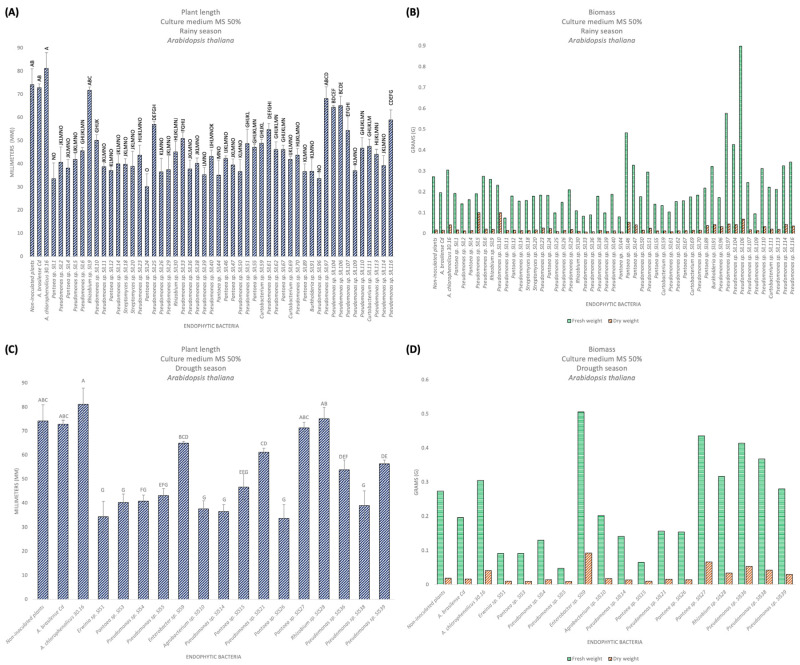
Promotion of plant growth of *A. thaliana* in MS 50% culture medium. (**A**) Length of the plant, rainy season. (**B**) Biomass in fresh and dry weight, rainy season. (**C**) Length of the plant, dry season. (**D**) Biomass in fresh and dry weight, dry season. The bars represent the mean value. Differences statistically significant with respect to the control bacteria were determined with a completely randomized ANOVA (Pr > F =< 0.0001), followed by a Tukey analysis (α = 0.05); the means with the same letter are not significantly different, and the error bars indicate the standard deviation of three repetitions. The biomass only represents the value of a replicate with 30 plants for each bacterial isolate.

**Figure 4 microorganisms-12-02654-f004:**
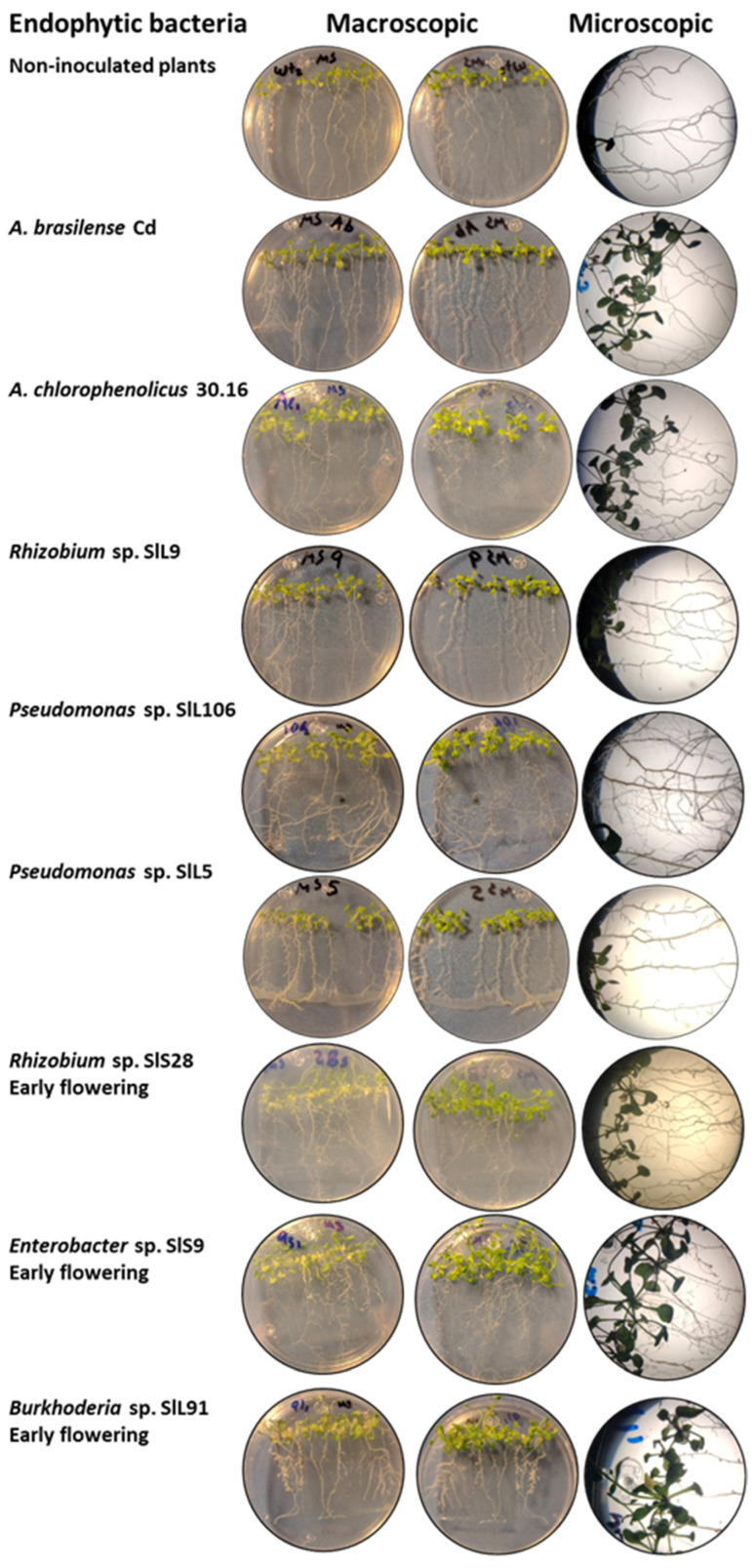
Appearance of *A. thaliana* plants inoculated with the different bacterial isolates in MS 50% culture medium.

**Figure 5 microorganisms-12-02654-f005:**
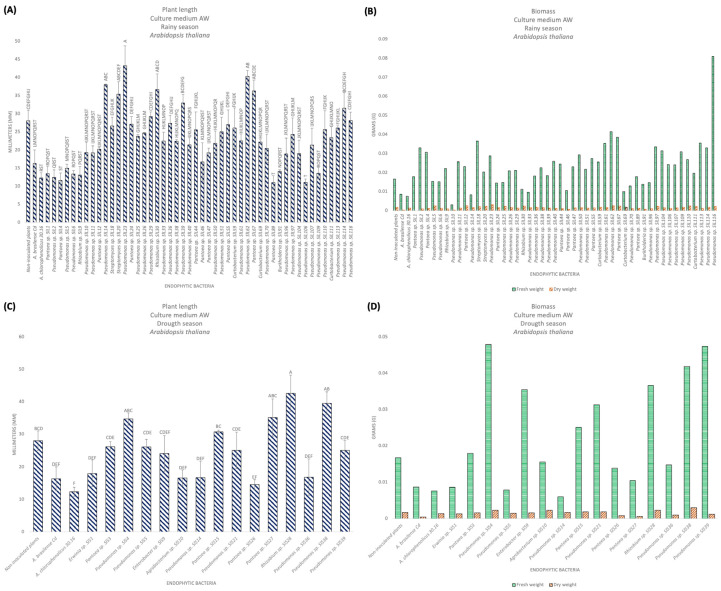
Promotion of plant growth of *A. thaliana* in AW culture medium. (**A**) Length of the plant, rainy season. (**B**) Biomass in fresh and dry weight, rainy season. (**C**) Length of the plant, dry season. (**D**) Biomass in fresh and dry weight, dry season. The bars represent the mean value. Differences that were statistically significant with respect to the control bacteria were determined with a completely randomized ANOVA (Pr > F = < 0.0001), followed by a Tukey analysis (α = 0.05); the means with the same letter are not significantly different, and the error bars indicate the standard deviation of three repetitions. The biomass represents the value of a replicate with 30 plants for each bacterial isolate.

**Figure 6 microorganisms-12-02654-f006:**
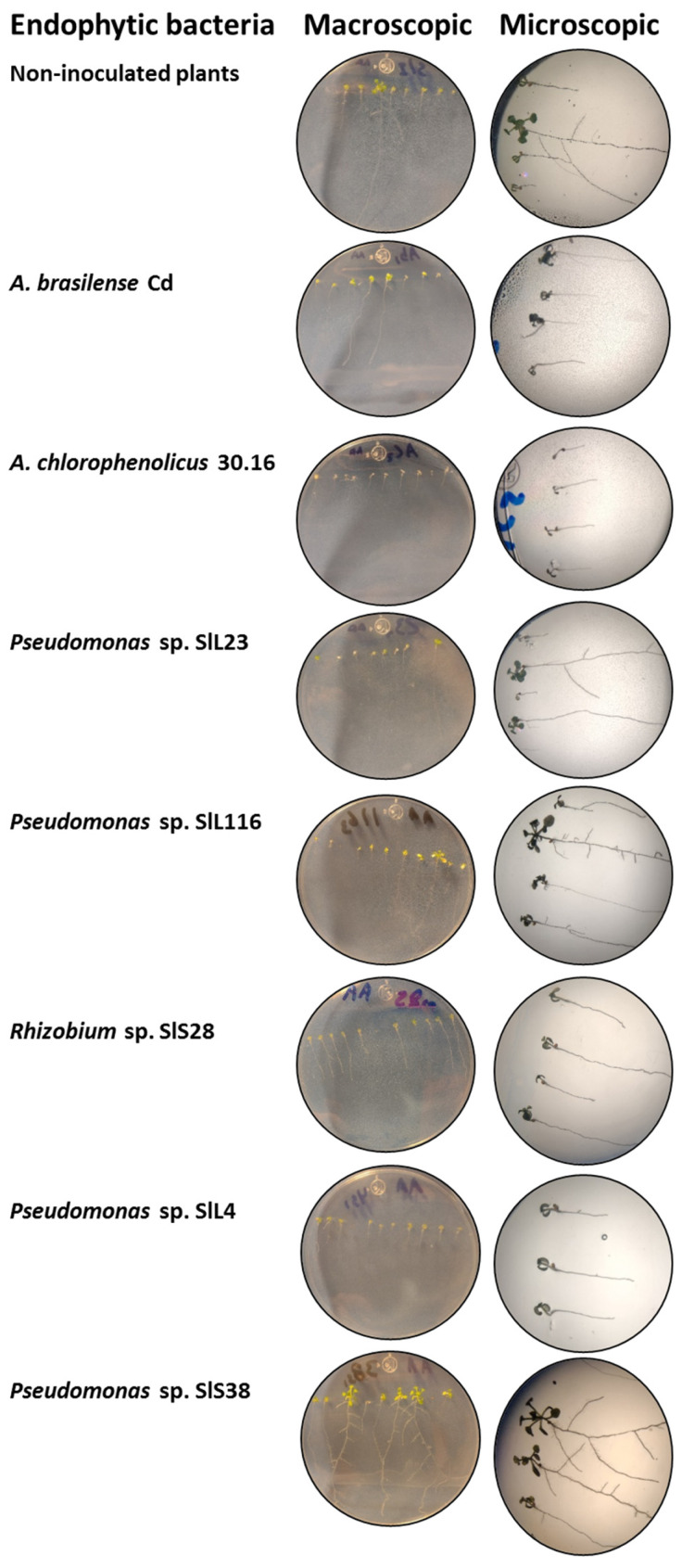
Appearance of *A. thaliana* plants inoculated with different bacterial isolates in AW culture medium.

**Figure 7 microorganisms-12-02654-f007:**
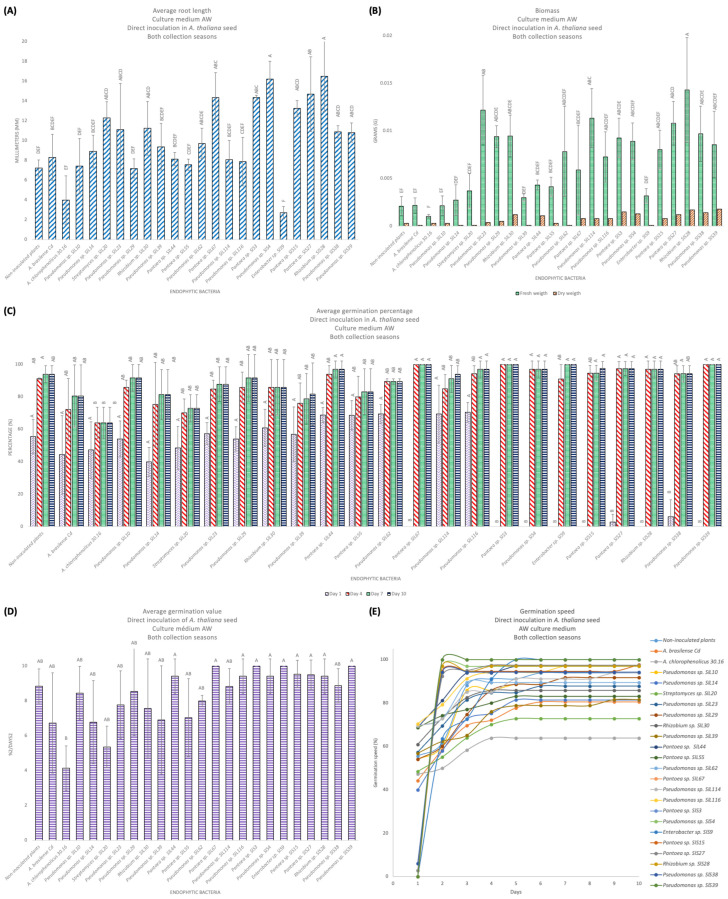
Promotion of plant growth of *A. thaliana* in AW culture medium. (**A**) Length of the plant, both seasons. (**B**) Biomass in fresh and dry weight, both seasons. (**C**) The germination percentage on days 1, 4, 7 and 10. (**D**) The germination speed on day 10. (**E**) The germination speed of *A. thaliana* seeds. The bars represent the mean value, and the statistically significant differences with respect to the control bacteria were determined with a completely randomized ANOVA (Pr > F = < 0.0001), followed by a Tukey analysis (α = 0.05); the means with the same letter are not significantly different, and the error bars indicate the standard deviation of three repetitions.

**Figure 8 microorganisms-12-02654-f008:**
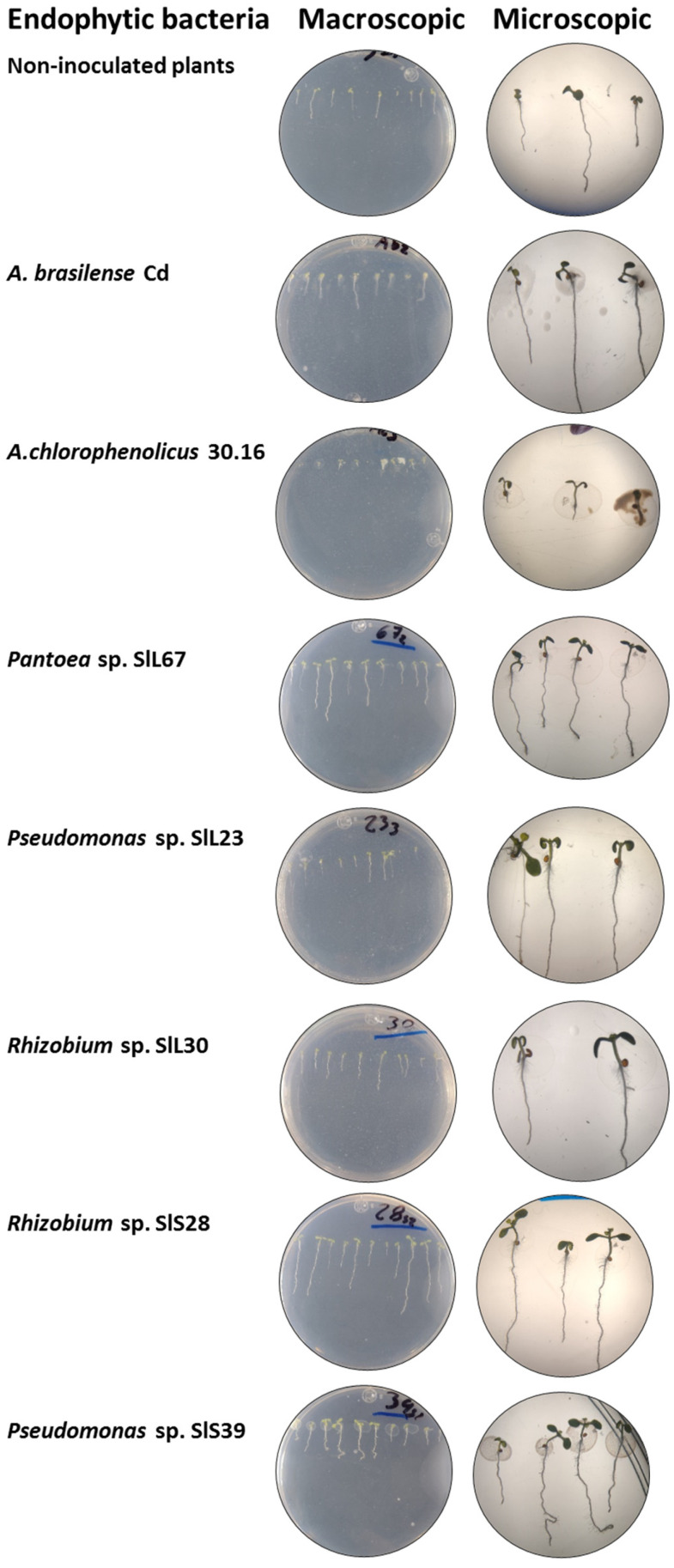
Appearance of *A. thaliana* plants inoculated directly on the seeds with different bacterial isolates in AW culture medium.

**Table 1 microorganisms-12-02654-t001:** Isolation of endophytic bacteria from *Selaginella lepidophylla* in hydrated and dehydrated states.

Hydrated State (Rainy Season)					
Culture Media	AN	LB	PY	PDA	Subtotal
Microphyll (Mi)	8	14	24	16	62
Rhizophore (Ri)	20	7	22	11	60
Subtotal	28	21	46	27	
Total	122				
**Dehydrated State (Drought Season)**					
Microphyll (Mi)	5	7	9	4	25
Rhizophore (Ri)	3	3	8	2	16
Subtotal	8	10	17	6	
Total	41				

**Table 2 microorganisms-12-02654-t002:** Frequency of bacterial isolates according to their IAA production levels.

	Without Tryptophan	With Tryptophan			
	48 h	24 h		48 h	
IAA Levels	Rain	Rain	Drought	Rain	Drought
Low producers (1.0 a 10.9 µg/mL)	0.81% (1)	69.01% (49)	32% (8)	62.37% (63)	13.33% (4)
Medium producers (11.0 a 20.9 µg/mL)	0	12.67% (9)	20% (5)	12.87% (13)	23.33% (7)
High producers (21.0 a 30.9 µg/mL)	0	18.30% (13)	48% (12)	24.75% (25)	63.33% (19)

**Table 3 microorganisms-12-02654-t003:** Plant growth-promotion parameters of endophytic bacteria from *Selaginella lepidophylla.* Plant growth promotion (PGP) in *A. thaliana* in MS 50% culture medium, AW culture medium, and in direct inoculation of the bacteria isolates in *A. thaliana* seeds in AW culture medium. Not determined (ND). According to the statistical analyses, significant differences were presented for the PGP experiments in MS 50% culture medium: F = 25.68 and F = 43.07 for the rainy and dry seasons, respectively, with a Pr > F < 0.0001; α = 0.05. For the experiments in AW culture medium: F = 21.36 and F = 16.43, for the rainy and dry seasons, respectively, with a Pr > F < 0.0001; α = 0.05. And in direct inoculation in *A. thaliana* with the bacterial isolates: F = 8.21 for both collection times; for the PF: F = 6.4 for both collection times, with a Pr > F < 0.0001; α = 0.05.

	PGP in *Arabidopsis thaliana* col. 0		
	MS 50%	AW	AW Direct Inoculation
Name	Root Length (mm)/Fresh Weight-Dry Weight (g)		
*A. brasilense* Cd	72.86/0.1966-0.0171	16.33/0.0087-0.0004	8.30/0.0021-0.0001
*A. chlorophenolicus* 30.16	81.17/0.305-0.0418	12.32/0.0076-0.0014	3.98/0.0010-0.0003
Bacteria isolated from *Selaginella lepidophylla* in the hydrated state (rainy season).			
*Pantoea* sp. SlL1	33.59/0.193-0.018	13.50/0.0179-0	ND
*Pseudomonas* sp. SlL2	40.71/0.1438-0.0139	12.42/0.0331-0.001	ND
*Pantoea* sp. SlL4	38.21/0.163-0.0159	11.61/0.0307-0.0012	ND
*Pseudomonas* sp. SlL5	42.01/0.1922-0.1009	14.95/0.0154-0.0026	ND
*Pseudomonas* sp. SlL6	45.68/0.2755-0.0223	13.39/0.0152-0.0013	ND
*Rhizobium* sp. SlL9	71.74/0.2611-0.0217	13.08/0.0222-0.0009	ND
*Pseudomonas* sp. SlL10	50.12/0.2334-0.1009	19.25/0.0031-0.0004	7.42/0.0021-0.0003
*Pseudomonas* sp. SlL11	38.76/0.0746-0.0154	19.23/0.0257-0.0014	ND
*Pantoea* sp. SlL12	37.20/0.1806-0.0174	20.23/0.0231-0.0027	ND
*Pseudomonas* sp. SlL14	40.02/0.1563-0.0149	38.01/0.0084-0.0016	8.92/0.0027-0.0001
*Streptomyces* sp. SlL18	39.82/0.1593-0.0146	26.63/0.0366-0.0021	ND
*Streptomyces* sp. SlL20	38.89/0.18-0.018	35.40/0.0204-0.0024	12.29/0.0037-0.0001
*Pseudomonas* sp. SlL23	43.73/0.1848-0.0267	43.21/0.0289-0.0032	11.10/0.0122-0.0004
*Pantoea* sp. SlL24	30.20/0.1844-0.0245	27.15/0.0145-0.0016	ND
*Pseudomonas* sp. SlL25	57.10/0.1-0.011	23.81/0.0148-0.0024	ND
*Pseudomonas* sp. SlL26	36.59/0.15-0.0154	24.71/0.021-0.0017	ND
*Pseudomonas* sp. SlL29	37.52/0.21-0.0201	29.24/0.0212-0.0014	7.17/0.0094-0.0005
*Rhizobium* sp. SlL30	45.18/0.11-0.0121	36.68/0.011-0.0025	11.24/0.0094-0.0012
*Pseudomonas* sp. SlL33	50.96/0.08-0.0078	22.47/0.009-0.0016	ND
*Pseudomonas* sp. SlL36	37.82/0.09-0.0091	27.37/0.018-0.0018	ND
*Pseudomonas* sp. SlL38	40.23/0.18-0.0158	22.41/0.022-0.0016	ND
*Pseudomonas* sp. SlL39	35.31/0.1-0.0103	32.95/0.0184-0.0014	9.37/0.003-0.0001
*Pseudomonas* sp. SlL40	43.14/0.19-0.0135	21.40/0.0259-0.0012	ND
*Pantoea* sp. SlL44	35.05/0.08-0.0084	25.55/0.0245-0.0011	8.12/0.0043-0.0011
*Pantoea* sp. SlL46	42.32/0.4843-0.0557	16.73/0.0106-0.0007	ND
*Pantoea* sp. SlL47	39.54/0.3279-0.0424	19.22/0.023-0.0015	ND
*Pseudomonas* sp. SlL50	36.69/0.1781-0.017	21.84/0.0294-0.0016	ND
*Pseudomonas* sp. SlL51	48.79/0.2952-0.0259	25.06/0.0217-0.0014	ND
*Pantoea* sp. SlL55	47.14/0.1418-0.0134	27.00/0.0274-0.0015	7.55/0.0041-0.0003
*Curtobacterium* sp. SlL59	48.92/0.1348-0.0138	26.05/0.0256-0.0018	ND
*Pseudomonas* sp. SlL61	54.75/0.1047-0.0098	22.55/0.0353-0.0017	ND
*Pseudomonas* sp. SlL62	46.17/0.1535-0.0138	40.32/0.0416-0.0027	9.71/0.0078-0.0001
*Pantoea* sp. SlL67	46.19/0.1576-0.0144	36.32/0.0386-0.0025	14.35/0.0059-0.0008
*Curtobacterium* sp. SlL69	41.93/0.177-0.0171	22.20/0.0101-0.0018	ND
*Pseudomonas* sp. SlL70	43.73/0.1852-0.0175	20.48/0.013-0.0011	ND
*Pantoea* sp. SlL89	36.67/0.2193-0.0391	11.02/0.0179-0.0013	ND
*Burkholderia* sp. SlL91	36.77/0.3219-0.0432	14.11/0.0139-0.0007	ND
*Pseudomonas* sp. SlL96	33.69/0.1729-0.0332	18.93/0.0147-0.0011	ND
*Pseudomonas* sp. SlL97	68.21/0.5771-0.0464	24.29/0.0336-0.0023	ND
*Pseudomonas* sp. SlL104	64.48/0.4275-0.043	19.01/0.0314-0.0016	ND
*Pseudomonas* sp. SlL106	65.08/0.9001-0.0698	11.06/0.0242-0.0013	ND
*Pseudomonas* sp. SlL107	54.38/0.245-0.0197	21.35/0.0241-0.0015	ND
*Pseudomonas* sp. SlL109	37.06/0.0956-0.0145	13.61/0.0309-0.0018	ND
*Pseudomonas* sp. SlL110	46.79/0.3134-0.0338	25.61/0.0268-0.0026	ND
*Curtobacterium* sp. SlL111	47.59/0.2235-0.0229	23.43/0.0197-0.002	ND
*Pseudomonas* sp. SlL113	44.09/0.2123-0.0209	26.06/0.0355-0.0013	ND
*Pseudomonas* sp. SlL114	39.20/0.351-0.0447	31.57/0.0331-0.0018	8.07/0.0113-0.0008
*Pseudomonas* sp. SlL116	59.03/0.3433-0.0366	28.10/0.0809-0.0022	7.87/0.0072-0.0008
Bacteria isolated from *Selaginella lepidophylla* in the dehydrated state (drought season).			
*Erwinia* sp. SlS1	34.32/0.0917-0.0104	17.95/0.0086-0.0013	ND
*Pantoea* sp. SlS3	40.24/0.0916-0.0096	26.14/0.0179-0.0016	14.34/0.0092-0.0015
*Pseudomonas* sp. SlS4	40.83/0.1301-0.0145	34.69/0.0479-0.0023	16.21/0.0089-0.0013
*Pseudomonas* sp. SlS5	43.16/0.0477-0.0095	26.06/0.0079-0.0015	ND
*Enterobacter* sp. SlS9	64.96/0.506-0.0933	24.05/0.0355-0.0016	2.71/0.0031-0.0001
*Agrobacterium* sp. SlS10	37.62/0.202-0.0182	16.59/0.0155-0.0023	ND
*Pseudomonas* sp. SlS14	36.52/0.1418-0.0144	16.70/0.006-0.0017	ND
*Pantoea* sp. SlS15	46.63/0.0653-0.01	30.73/0.0251-0.0019	13.23/0.0080-0.0008
*Pseudomonas* sp. SlS21	61.14/0.1574-0.016	25.05/0.0313-0.0019	ND
*Pantoea* sp. SlS26	33.78/0.1544-0.0145	14.54/0.0139-0.0008	ND
*Pantoea* sp. SlS27	71.28/0.4356-0.0671	35.14/0.0105-0.0006	14.68/0.0108-0.0012
*Rhizobium* sp. SlS28	75.13/0.3169-0.0344	42.63/0.0366-0.0023	16.48/0.0143-0.0017
*Pseudomonas* sp. SlS36	53.82/0.4146-0.0537	16.83/0.0148-0.001	ND
*Pseudomonas* sp. SlS38	38.95/0.3679-0.043	39.54/0.0419-0.003	10.87/0.0097-0.0014
*Pseudomonas* sp. SlS39	56.37/0.28-0.0303	25.04/0.0474-0.0012	10.81/0.0085-0.0018

## Data Availability

The accession numbers of the 16S rRNA gene sequence deposited in the NCBI GenBank are registered under the following accession numbers: PQ570561-PQ570562, PQ541125-PQ541126, and PQ497908-PQ497967. However, they are under embargo until the article is published.

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
