# Peer review of "Endophytic Bacteria from the Desiccation-Tolerant Plant Selaginella lepidophylla and Their Potential as Plant Growth-Promoting Microorganisms"

_microorganisms, 2024, doi:10.3390/microorganisms12122654_

Round 1

Reviewer 1 Report

Comments and Suggestions for Authors

It is a very nice and informative study. The review report is attached.

Author Response

RESPONSE TO REVIEWER 1

  1. Introduction: There is still a long way to go from isolating plant growth-promoting bacteria to practical application, so it is recommended to add a paragraph regarding the potential of endophytes in agricultural applications.

Response:

The wide diversity of PGPB in agricultural systems suggests the elemental role they play in plant physiology and functioning. Studies have shown how beneficial they are in agricultural crops in relation to their growth, development and adaptability to biotic and abiotic stress due to the diversity of their PGP mechanisms. However, more research needs to be conducted on the various PGPBs that produce synergistic effects and their application for PGP.

  1. Introduction: Please clearly state the research gaps in this study.

Response:

This study aims to isolate, characterize, and identify plant growth promoting endophytic bacteria of S. lepidophylla from its natural environment during the rainy and dry seasons. In addition, to select the ideal candidates to evaluate the promotion of plant growth in Arabidopsis thaliana plants with the endosymbionts of S. lepidophylla. This will allow us to determine the role of the endophytic microbiota associated with the resurrection plant S. lepidophylla to try to understand the role or physiological and molecular contribution of endophytic bacteria to the plant.

  1. Materials and Methods: L166 2.5. Analysis of indole acetic acid (IAA) production. The Salkowski colorimetric technique is a commonly used method to evaluate a strain's capacity to produce the phytohormone indole acetic acid (IAA). However, Glickmann and Dessaux (1995) demonstrated that the Salkowski reagent also reacts positively with other indoles, such as indole pyruvic acid and indole acetamide. Consequently, the Salkowski test detects the presence of all indole-like molecules produced by a bacterial strain, rather than being specific to IAA. (de-Bashan and Nannipieri, 2024). Please mention the limitations and possible effect of this method when discussing the IAA. Glickmann E, Dessaux Y (1995) A Critical examination of the specificity of the Salkowski reagent for indolic compounds produced by phytopathogenic bacteria. Appl Environ Microbio 61:793–796 de-Bashan, L. and Nannipieri, P., 2024. Recommendations for plant growth-promoting bacteria inoculation studies. Biology and Fertility of Soils, 60(3), pp.259-261.

Response:

Although the Salkowski colorimetric technique is the standard method for determining the ability of bacteria to produce IAA. This technique not only detects IAA but also detects other indoles such as indole pyruvic acid and indole acetamide. To accurately determine IAA production, high-performance liquid chromatography analysis should be used. However, Salkowsi's method is widely used and valid (de-Bashan, L., & Nannipieri, 2024).

de-Bashan, L., & Nannipieri, P. (2024). Recommendations for plant growth-promoting bacteria inoculation studies. Biology and Fertility of Soils, 60(3), 259-261. https://doi.org/10.1007/s00374-024-01798-w

  1. Results: Figure 3, 5 and 7 needs to be redrawn so that readers can distinguish the letters above the bars.

Response:

Figures 3, 5 and 7 were edited and improved.

  1. Discussion: The authors mentioned the prospect of agricultural application in the discussion, but did not conduct an in-depth analysis of the differences between the results under laboratory conditions and actual field conditions. It is recommended to supplement the discussion on the feasibility and technical challenges of future field validation experiments.

Response

The next paragraph was added to the discussion:

The application of PGPB as a biofertilizer is an encouraging strategy to enhance the efficiency of agricultural cultivation. However, the efficiency of PGPB depends largely on extrinsic and intrinsic factors such as plant genetics, environmental conditions and other specific characteristics of agricultural systems.

Reviewer 2 Report

Comments and Suggestions for Authors

The manuscript presents interesting data on a very current topic in the field of agro-food biotechnology. The experiments are well designed. Perhaps some experiments could have been carried out with a strain in a plant of agricultural interest to see if the results of Arabidopsis under such controlled conditions could be extrapolated to other less controlled conditions. All the results and decisions are based on biometric measurements, without making any physiological assessment of the plants. I understand that the results are presented in a journal with a defined scope and that aspects of plant physiology may not fit well, but they would certainly have greatly reinforced the conclusions of the work. Having said that, I think that the work can be published, but first changes would necessarily have to be made in two aspects.

Figures 3, 5 and 7 cannot be presented like this. Nothing is visible, the letters of the bars overlap each other and the axes are very small. The letters of the bars are usually presented in lowercase, but that is the least of the problems. I don't know if the solution would be to lengthen the figures and make them individual instead of A, B, C... or to make them tables, but as they are, no.

Another aspect to improve is the discussion. It is too long and in many places too speculative. As an example, lines 546-572. At one point in this paragraph the authors say: "The abundance of endophytic bacteria during the rainy season could be related to their participation in stress tolerance or their future participation......This is almost a fortune teller's prediction.

Author Response

RESPONSE TO REVIEWER 2

  1. The manuscript presents interesting data on a very current topic in the field of agro-food biotechnology. The experiments are well designed. Perhaps some experiments could have been carried out with a strain in a plant of agricultural interest to see if the results of Arabidopsis under such controlled conditions could be extrapolated to other less controlled conditions. All the results and decisions are based on biometric measurements, without making any physiological assessment of the plants. I understand that the results are presented in a journal with a defined scope and that aspects of plant physiology may not fit well, but they would certainly have greatly reinforced the conclusions of the work. Having said that, I think that the work can be published, but first changes would necessarily have to be made in two aspects.

Response:

The model plant A. thaliana ecotype Columbia 0 was used to evaluate the PGP of our endophytic bacteria, because this plant has a short life cycle, grows in limited spaces, presents natural variations (ecotypes) and in addition to being counted with various transgenic and mutant lines that can be used in the future to evaluate other capacities of endophytic bacteria. Furthermore, as a perspective we plan to carry out experiments on the plant of agricultural interest Solanum lycopersicum.

  1. Figures 3, 5 and 7 cannot be presented like this. Nothing is visible, the letters of the bars overlap each other and the axes are very small. The letters of the bars are usually presented in lowercase, but that is the least of the problems. I don't know if the solution would be to lengthen the figures and make them individual instead of A, B, C... or to make them tables, but as they are, no.

Response:

Figures 3, 5 and 7 were edited and improved.

  1. Another aspect to improve is the discussion. It is too long and in many places too speculative. As an example, lines 546-572. At one point in this paragraph the authors say: "The abundance of endophytic bacteria during the rainy season could be related to their participation in stress tolerance or their future participation......This is almost a fortune teller's prediction.

Response:

We have shortened the discussion and eliminated those speculative phrases

Reviewer 3 Report

Comments and Suggestions for Authors

Material and Methods

2.2. Plant material sterilization and bacterial endophytes isolation

To disinfest the explants, 10% commercial chlorine or 20% commercial chlorine was used, but the concentration of commercial chlorine is not clear because the disposition in the commercial sodium hypochlorite solution is 10 to 15% chlorine.

Results and Discussion

Justify why Arabidopsis thaliana was chosen for the growth promotion trial and not a plant with nutritional or medicinal interest, so that the study has an application in the production of plants of agricultural importance. Auxin, a plant hormone, regulates virtually every aspect of plant growth and development. Many current studies on auxin focus on the model plant Arabidopsis thaliana, or on field crops, such as rice and wheat. There are relatively few studies on what role auxin plays in various physiological processes of a range of horticultural plants (Zhang Q, Gong M, Xu X, Li H, Deng W. Roles of Auxin in the Growth, Development, and Stress Tolerance of Horticultural Plants. Cells. 2022 Sep 5;11(17):2761. doi: 10.3390/cells11172761.)

Author Response

RESPONSE TO REVIEWER 3

  1. Plant material sterilization and bacterial endophytes isolation

To disinfest the explants, 10% commercial chlorine or 20% commercial chlorine was used, but the concentration of commercial chlorine is not clear because the disposition in the commercial sodium hypochlorite solution is 10 to 15% chlorine.

Response:

Commercial chlorine from Cloralex® brand was used, which has a concentration of 6.15% sodium hypochlorite.

  1. Results and Discussion: Justify why Arabidopsis thaliana was chosen for the growth promotion trial and not a plant with nutritional or medicinal interest, so that the study has an application in the production of plants of agricultural importance. Auxin, a plant hormone, regulates virtually every aspect of plant growth and development. Many current studies on auxin focus on the model plant Arabidopsis thaliana, or on field crops, such as rice and wheat. There are relatively few studies on what role auxin plays in various physiological processes of a range of horticultural plants (Zhang Q, Gong M, Xu X, Li H, Deng W. Roles of Auxin in the Growth, Development, and Stress Tolerance of Horticultural Plants. Cells. 2022 Sep 5;11(17):2761. doi: 10.3390/cells11172761.)

Response:

The model plant A. thaliana ecotype Columbia 0 was used to evaluate the PGP of our endophytic bacteria, because this plant has a short life cycle, grows in limited spaces, presents natural variations (ecotypes) and in addition to being counted with various transgenic and mutant lines that can be used in the future to evaluate other capacities of endophytic bacteria. Furthermore, as a perspective we plan to carry out experiments on the plant of agricultural interest Solanum lycopersicum.

Reviewer 4 Report

Comments and Suggestions for Authors

The authors present a simple and interesting study of the plant growth-promoting potential of the large selection of isolates from Selaginella lepidophylla. At the beginning of publication, authors provide a gripping introduction to the topic explaining the main idea of the manuscript. The selected simple methods are perfectly chosen to verify scientific problems and are explained clearly. For the result section I would suggest using a different formula for the analysis of phosphate solubilization and siderophore production as the units in the currently used one do not add up. Additionally, the currently used graphs are hard to read, and a larger version of these graphics should be uploaded. The publication ends with a long and insightful discussion summarizing the obtained interesting results. In my opinion, the strength of this article lies within the activity of strains isolated from plants both from dry and wet seasons and using different culture media, there is not much data published that presents this kind of data. Additionally, the used methods allowed the authors to perform a wide analysis of the isolated endophytes which can be used to conclude the microbial communities of Selaginella lepidophylla during the dry and wet seasons. In my opinion, this is an excellent manuscript deserving of publication in Microorganisms after minor revisions.

Supplementary table 1 please add the accession numbers for the ERIC controls with appropriate citations. Please indicate what the yellow highlighting means in Sil67

Line 107: salicylic acid is volatile in the form of its ester

Line 136: It is Lysogenic Agar (LA), not Luria Bertani (LB), it is a common misunderstanding, but its creator Giuseppe Bertani named it Lysogenic Agar.

Line 168: Please indicate the shaking rpm’s

Line 178: Please add information about the negative control.

Line 186: It is unclear to me why this formula is used, the colony size informs us how well the strains grow on the medium with insoluble phosphate and therefore can be an indication of the bacteria's availability to solubilize it, and the second part is informing us how well the strain solubilizes phosphate in the medium but is without unit (the mm from the halo radius cross out the mm from the colony radius or diameter). Please do not add values with different units. For me, the ratio or the halo size are both reasonable measurements, but please choose one or you can use both just not their sum.

Line 210: Please use rcf instead of rpm,

Line 221: It is a strange formula and units do not add up. Additionally please indicate what was measured diameter or radius.

Line 224: washed with water and sterilized with…

Line 330: It is interesting results

Line 408: Please upload a higher resolution version of this figure, especially the letters indicating statistical differences are not visible in panel A, the names of strains are not fully displayed and no strain number is visible.

Line 445: As mentioned above, please use larger figures, and make sure all information is visible.

Line 823: Please fill in the Data availability statement

Author Response

RESPONSE TO REVIEWER 4

  1. The authors present a simple and interesting study of the plant growth-promoting potential of the large selection of isolates from Selaginella lepidophylla. At the beginning of publication, authors provide a gripping introduction to the topic explaining the main idea of the manuscript. The selected simple methods are perfectly chosen to verify scientific problems and are explained clearly. For the result section I would suggest using a different formula for the analysis of phosphate solubilization and siderophore production as the units in the currently used one do not add up. Additionally, the currently used graphs are hard to read, and a larger version of these graphics should be uploaded. The publication ends with a long and insightful discussion summarizing the obtained interesting results. In my opinion, the strength of this article lies within the activity of strains isolated from plants both from dry and wet seasons and using different culture media, there is not much data published that presents this kind of data. Additionally, the used methods allowed the authors to perform a wide analysis of the isolated endophytes which can be used to conclude the microbial communities of Selaginella lepidophylla during the dry and wet seasons. In my opinion, this is an excellent manuscript deserving of publication in Microorganisms after minor revisions.

Response:

Figures were edited and improved. The data was corrected.

  1. Supplementary table 1 please add the accession numbers for the ERIC controls with appropriate citations. Please indicate what the yellow highlighting means in Sil67

Response:

  1. brasilense Cd, A. chlorophenolicus 30.16 and P. lutea strains were used as reference only in the IAA quantification, phosphate solubilization, biological nitrogen fixation and siderophore synthesis assays; but not by the ERIC-PCR.

Yellow highlighting in SlL67 does not indicate anything, it was a mistake, but it has already been corrected.

  1. Line 107: salicylic acid is volatile in the form of its ester

Response:

The data was corrected.

  1. Line 136: It is Lysogenic Agar (LA), not Luria Bertani (LB), it is a common misunderstanding, but its creator Giuseppe Bertani named it Lysogenic Agar.

Response:

Accuracy is correct and acronyms have been corrected throughout the manuscript.

  1. Line 168: Please indicate the shaking rpm’s

Response:

Shaking is 200 rpm

  1. Line 178: Please add information about the negative control.

Response:

There was no negative control.

  1. Line 186: It is unclear to me why this formula is used, the colony size informs us how well the strains grow on the medium with insoluble phosphate and therefore can be an indication of the bacteria's availability to solubilize it, and the second part is informing us how well the strain solubilizes phosphate in the medium but is without unit (the mm from the halo radius cross out the mm from the colony radius or diameter). Please do not add values with different units. For me, the ratio or the halo size are both reasonable measurements, but please choose one or you can use both just not their sum.

Response:

The plates were incubated at 30 °C for 7 days. A clear zone around a growing colony indicated phosphate solubilization and was measured as phosphate solubilization index (SI). SI was calculated as the diameter of the total diameter (colony + halo zone) to the colony diameter.

Pande, A., Pandey, P., Mehra, S., Singh, M., & Kaushik, S. (2017). Phenotypic and genotypic characterization of phosphate solubilizing bacteria and their efficiency on the growth of maize. Journal of Genetic Engineering and Biotechnology, 15(2), 379-391. https://doi.org/10.1016/j.jgeb.2017.06.005

  1. Line 210: Please use rcf instead of rpm,

Response:

The data was corrected.

  1. Line 221: It is a strange formula and units do not add up. Additionally please indicate what was measured diameter or radius.

Response:

The data was corrected.

  1. Line 224: washed with water and sterilized with…

Response:

Seeds of A. thaliana ecotype Col. 0, were rinsed with water and sterilized with 20% NaClO and 0.2% tween 20 for 5 min.

  1. Line 330: It is interesting results
  2. Line 408: Please upload a higher resolution version of this figure, especially the letters indicating statistical differences are not visible in panel A, the names of strains are not fully displayed and no strain number is visible.

Response:

Figure was edited and improved

  1. Line 445: As mentioned above, please use larger figures, and make sure all information is visible.

Response:

Figures were edited and improved.

  1. Line 823: Please fill in the Data availability statement

Response:

Data was incorporated.

Reviewer 5 Report

Comments and Suggestions for Authors

Ms. "Endophytic bacteria from the dehydration-tolerant plant Selaginella lepidophylla and their potential as plant growth-promoting microorganisms" is well written. The topic is interesting and relevant to the readers.

Introduction: Introduces the reader in the context of the topic studied. However, the whole section is slightly chaotically written. It needs to bring organization, separate paragraphs, highlight previous studies and its own objectives.

 Materials and Methods: The authors describe in detail the type of research they conducted, how they collected and analyzed the data and the instruments they used in the research.

Results: Findings are reported objectively.

Figures 3, 5 even 7 should be split into two to be visible.

 Discussion: The first paragraph consists of a brief review of the main findings of the study.

541: add  a reference at least.

546-548, 549-550, 555 need a a reference each. 

Basically, the references given at the end of each paragraph should be redistributed within it. This style is confusing.

Finally, what are the limitations of the study?

Conclusions: are succinct. Future studies are underlined.

Author Response

RESPONSE TO REVIEWER 5

  1. Ms. "Endophytic bacteria from the dehydration-tolerant plant Selaginella lepidophylla and their potential as plant growth-promoting microorganisms" is well written. The topic is interesting and relevant to the readers.
  2. Introduction: Introduces the reader in the context of the topic studied. However, the whole section is slightly chaotically written. It needs to bring organization, separate paragraphs, highlight previous studies and its own objectives.

Response:

The introduction has been rearranged

  1. Materials and Methods: The authors describe in detail the type of research they conducted, how they collected and analyzed the data and the instruments they used in the research.
  2. Results: Findings are reported objectively.
  3. Figures 3, 5 even 7 should be split into two to be visible.

Response:

Figures 3, 5 and 7 were edited and improved.

  1. Discussion: The first paragraph consists of a brief review of the main findings of the study.
  2. 541: add a reference at least.

Response:

References were included.

  1. 546-548, 549-550, 555 need a a reference each. Basically, the references given at the end of each paragraph should be redistributed within it. This style is confusing.

Response:

References were included and incorporated at the end of the paragraph.

  1. Finally, what are the limitations of the study?

Response:

Some limitations of this work are that only the endophytic bacterial microorganisms that were culturable were obtained; likewise, in this work only a limited number of endo-phytic bacteria were identified. Omics tools such as metagenomics, must be used to un-derstand the wide diversity of microorganisms that Selaginella harbors.

  1. Conclusions: are succinct. Future studies are underlined.

Response:

Conclusion was edited and improved.

Round 2

Reviewer 2 Report

Comments and Suggestions for Authors

The authors have made the suggested changes and the manuscript has improved. I must say, however, that in response to my comment: "the results and decisions are based on biometric measurements, without making any physiological assessment of the plants", the authors have avoided answering directly and have made a comment to get out of the way.